

# The influence of albedo parameterization for improved lake ice simulation

Alexis L. Robinson[1], Sarah S. Ariano[2], Laura C. Brown[1]

[1]Department of Geography, University of Toronto Mississauga, Mississauga, L5L 1C6, Canada
[2]Department of Geography and Environmental Studies, Ryerson University, Toronto, M5B 1G3, Canada

*Correspondence to:* Alexis L. Robinson (alexis.robinson@mail.utoronto.ca)

**Abstract.** Lake ice models can be used to study the latitudinal differences of current and projected changes in ice covered lakes under a changing climate. The Canadian Lake Ice Model (CLIMo) is a one-dimensional freshwater ice cover model that simulates Arctic and sub-Arctic lake ice cover well. Modelling ice cover in temperate regions has presented challenges due to
the differences in composition between northern and temperate ice. This study presents a comparison of measured and modelled ice regimes, with a focus on refining CLIMo for temperate regions. The study sites include two temperate region lakes (MacDonald Lake and Clear Lake, Central Ontario) and two High Arctic lakes (Resolute Lake and Small Lake, Nunavut) where climate and ice cover information have been recorded over three seasons. The ice cover simulations were validated with a combination of time lapse imagery, field measurements of snow depth, snow density, ice thickness and albedo data, and
historical ice records from the Canadian Ice Database (for Resolute Lake). Simulations of the High Arctic ice cover show good agreement with previous studies for ice-on and ice-off dates (MAE 6 to 8 days). Unadjusted simulations for the temperate region lakes show both an underestimation in ice thickness (~ 4 to 18 cm) and ice-off timing (~ 25 to 30 days). Field measurements were used to adjust the albedo parameterization used in CLIMo, which resulted in improvements to both simulated ice thickness, within 0.1 cm to 10 cm of manual measurements, and ice-off timing, within 1 to 7 days of observations.
These findings suggest regionally specific measurements of albedo can improve the accuracy of lake ice simulations. These results further our knowledge regarding of the response of temperate and High Arctic lake ice regimes to climate conditions.

## 1   Introduction

Globally, the greatest spatial distribution of freshwater lakes is between 45–75°N (Verpoorter et al., 2014) with most of these
lakes experiencing some level of ice cover throughout the year. Reported trends in lake ice cover have shown shifts towards shorter ice-covered seasons, with the rates of change depending on the time span examined (e.g. Benson et al., 2012). Long-term trends of Northern Hemisphere lakes project the number of lakes transitioning from annual ice cover to intermittent winter ice cover will increase exponentially with climate warming (Sharma et al., 2019). Regional water and energy balances will likely experience changes as a result of this decreasing ice cover; through changes to the exchange of moisture and gas fluxes





(e.g. increased evaporation), as well as to ecosystems (e.g. earlier lake stratification, warmer summer surface temperatures and
increased aquatic productivity) and socio-economic changes (e.g. reduced winter recreation and transportation) (Brown and
Duguay, 2010; Benson et al., 2012; Arp et al., 2015; Duguay et al., 2015; Griffiths et al., 2017; Hampton et al., 2017; Hewitt
et al., 2018). These changes and their impacts vary spatially, due to latitudinal differences in ice types and how they respond
to climate. With northern latitudes warming at a more rapid rate than southern latitudes (AMAP, 2017), the latitudinal response

of lake ice becomes even more pertinent.

       The dominant controls on ice phenology (ice-off, ice-on, duration) are lake size, air temperature, precipitation, wind
speed, and the radiation balance (Brown and Duguay, 2010; Leppäranta, 2015). After formation, ice thickens as a result of
conductive heat loss from the warmer water below through the ice/snow cover to the atmosphere, leading to the formation of
black ice (Leppäranta, 2015). On-ice snow accumulation can both modify the thickness through insulating properties and

contribute to thickening from flooded snow or slush refreezing into the ice sheet as white ice (Brown and Duguay, 2010;
Leppäranta, 2015). Additionally, in temperate regions ice thickening through white ice growth has been observed to occur
through multiple mid-winter freeze/thaw events (Ariano and Brown, 2019). The albedo of the ice and any overlying snow
plays a large role in the ice-off process as ice melt initiation is controlled by albedo, which is a surface property describing the
ratio of outgoing to incoming solar radiation (Duguay et al., 2003). Lake ice albedo is affected by snow cover, ice type (e.g.

black ice and white ice), ice thickness, the presence of impurities, cloud cover, air temperature and solar zenith angle
(Leppäranta, 2015). The light scattering properties of white ice are different than black ice due to the tightly packed air bubbles,
which result in a higher albedo (Svacina et al., 2014a; Leppäranta, 2015). Both snow cover and white ice delay ice-off due to
the higher albedo compared to the lower albedo of black ice (Svacina et al., 2014a; Leppäranta, 2015), however, when snow-
free conditions exist on the ice or moisture is present in the white ice, albedo begins to decrease, and melt occurs. Once snow

and ice melt occur, albedo values drop to between 0.7 and 0.25; which results in more energy available for melt and creates a
positive feedback loop that accelerates the melt process (Heron and Woo, 1994; Henneman and Stefan, 1999; Jakkila et al.,
2009; Leppäranta, 2015; Zdorovennova et al., 2018). Studies of lake ice albedo indicate that values can range from 0.10 - 0.85
(Heron and Woo, 1994; Henneman and Stefan, 1999; Jakkila et al., 2009; Semmler et al., 2012; Svacina et al., 2014b;
Zdorovennova et al., 2018). In addition, the albedo measured for snow on ice covered lakes ranges between 0.5 ¬ 0.95 (Jakkila

et al., 2009; Semmler et al., 2012; Svacina et al., 2014a; Zdorovennova et al., 2018). These studies indicate that lake ice (and
overlying snow) albedo values vary temporally and spatially which makes the parameterization of albedo in modelling
applications difficult (Lang et al., 2018; Leppäranta, 2015), highlighting the need to better understand and improve the
representation of albedo for ice regimes within lake ice models.

       Physically based models can be used to simulate ice phenology thickness and to examine the sensitivity of these

factors to climate change. (e.g. Brown and Duguay, 2011a; Yang et al., 2012; Cheng et al., 2014). The Canadian Lake Ice
Model (CLIMo) has been used successfully for simulating Arctic and sub-Arctic lake ice cover (e.g. Ménard et al., 2002;
Duguay et al., 2003; Morris et al., 2005; Brown and Duguay, 2011a, 2011b; Kheyrollah Pour et al., 2012; Surdu et al., 2014).
This model uses a thermodynamic approach to determine ice formation, growth and decay; where these processes are





controlled by an energy surplus or deficit (Brown and Duguay, 2010). However, CLIMo differs from other thermodynamic

models through its parameterization of snow conductivity and surface albedo (Duguay et al., 2003). The albedo parameterization in CLIMo is dependent on surface type (ice, snow, or open water), whether the surface temperature is above or below freezing, and the thickness of the ice (Duguay et al., 2003). Currently, albedo is parametrized following Maykut, (1982) for cold conditions and uses observation data from High Arctic lake ice (Heron and Woo, 1994) for melting conditions. These observations were obtained from Small Lake, NU, where the typical ice conditions exceed two metres with a very small

amount (< 1%–7%, Heron, 1985) of white ice formed at the top of the ice column (white ice was measured at < 4% of the total ice thickness; Heron, 1985; Ariano and Brown, 2019). In the temperate regions, however, typical ice conditions do not exceed 1 m and have a much higher amount of white ice. For example, in the Haliburton region of Central Ontario, maximum white ice percentages ranged from 25–73% of the ice column between 2016 and 2019 (Ariano and Brown, 2019, updated to 2019). Initial model simulations for this region using CLIMo found that ice-off dates were too early, as the large amount of white ice

was not accounted for (Ariano, 2017), indicating that the current parameterization of surface albedo in CLIMo is not suitable for temperate lake ice.

This research compares ice cover simulations from High Arctic and temperate region lakes to illustrate the latitudinal differences in lake ice properties and presents refinements to CLIMo to better simulate ice thickness and ice-off timing in the temperate region. The specific objectives of this research were to 1) show the effectiveness of CLIMo for simulating the ice

cover regimes of a small (< 1 km$^2$) and a medium (1–100 km$^2$) sized High Arctic lakes and 2) investigate and improve the ability of CLIMo to simulate temperate region ice covers using two medium sized lakes in Central Ontario. Understanding the latitudinal differences in lake ice processes, types, and the interactions of climate and lake ice is important for improving climate and lake ice modelling accuracy – which is an essential precursor to providing more robust modelling results of predicted changes.

**2      Study Areas**

**2.1      Study Area: High Arctic**

Small Lake and Resolute Lake (Fig. 1a, b, c) are located on Cornwallis Island in Nunavut, Canada. Small Lake (74°45'N, 95°05'W) has a surface area of 0.2 km2 (Woo, 2012) and a maximum depth of ~10 m (Heron and Woo, 1994). Ice-off and ice-on, recorded by digital cameras from 2016-2018, occurred late July to early August and early September to late September,

respectively. Maximum ice thickness in 1980 and 1981 was 2.42 m ± 0.1 m and 2.37 m ± 0.1 m, respectively (Heron, 1985) and in May 2016 it was measured at ~1.88 m with the ice type being composed of almost entirely black ice (Brown, 2016).

Resolute Lake (74.72°N, 94.95°W) has a surface area of 1.27 km2 and a maximum depth of 22.5 m (Lescord et al., 2015). Historical ice date records between 1961–1986 for Resolute Lake are recorded in the Canadian Ice Database (CID) (Lenormand et al., 2002), where ice-off and ice-on occurred late July to late August and early September to early October,

respectively. Mean maximum ice thickness measured in June between 1970 to 1982 was ~2.13 m (Lenormand et al., 2002).





Recently, thickness was measured on May 19, 2019 and ranged from 2.1 to 2.4 m (Data collected by Resolute community members).

The location of the Resolute Bay weather station (74°43'N, 94°58'W) is ~ 4.9 km south-east of Small Lake and ~3 km north of Resolute Lake. The mean climatology (1981-2010) (ECCC, 2017) indicates subfreezing temperatures last for 9 months (from September to May) with a mean winter temperature of -31.5 °C and a mean summer temperature of 2.3 °C. Mean snowfall is 111.2 cm; with a mean end of May snow depth of 17 cm (ECCC, 2017) (used to represent end-of-season snow). However, the representativeness of weather station snowfall to snow accumulation in High Arctic basins has been shown to be under-represented by 130-300% (Woo et al., 1983; Yang and Woo, 1999). End of season snow surveys on Small Lake (Table 1) completed May 22, 2016 and May 16, 2018 measured mean snow depth on the lake at 17 cm and 11 cm, respectively. In 1980 and 1981 snow surveys were completed throughout June, with the central part of the lake mean snow depth measuring 0.1 m or less and the mean snow depth at edges of the lake measuring 1.5 m or less (Heron, 1985). Finally, the mean snow density for May, extracted from the Canadian Snow Database CD (Snow CD) (which contains gridded snow density normals between 1960 to 1990, Meteorological Service of Canada (MSC), 2000), for the region is 303 kg m$^{-3}$, while the measured on-ice snow density from the snow surveys was 357 kg m$^{-3}$ in 2016 and 308 kg m$^{-3}$ in 2018; slightly higher than on land, as on-ice snow densities are typically higher by ~20 % (Sturm and Liston, 2003).

### 2.2 Study Area: Temperate

The temperate study lakes, MacDonald Lake and Clear Lake, are located within the Haliburton Forest and Wild Life Reserve Ltd. (45.12° N, 78.07° W) in Central Ontario, Canada (Fig. 1 d, e). Haliburton County is located at the southern end of the Precambrian shield, in the Ontario Shield ecoregion, and is defined by a temperate climate and dominated by mixed and deciduous forests (Crins et al., 2009; Hadley et al., 2013). MacDonald Lake has a surface area of 1.56 km$^2$ and a maximum depth of 39.6 m (mean depth 12.2 m) while the surface area of Clear Lake is 1.8 km$^2$ with a maximum depth of 42.7 m (mean depth 15.2 m) (Haliburton Forest and Wild Life Reserve, 2012).

The mean climatology records (1981–2010) (ECCC, 2017) from the nearby Town of Haliburton, Ontario (45.03 °N, 78.53 °W; 22 km south of the study lakes) show sub-freezing temperatures last for 4 months (December to March) with a mean winter temperature of -5.0 °C and a mean summer temperature of 16.6 °C. The mean annual snowfall is 279.6 cm with a mean end of March snow depth of 16 cm (ECCC, 2017). On-ice mean snow depth and density for MacDonald Lake and Clear Lake were collected over three field seasons (2016, 2017, and 2018; Table 2) and showed considerable variability throughout all three field seasons. The mean snow depth on the lakes in 2016 was 14 cm; in 2017 it was 12 cm; and in 2018 it was 7 cm. The range of snow densities (Table 2) provided from the Canadian Snow CD (MSC, 2000) indicate mean snow densities of 259 kg m$^{-3}$. In comparison, mean field snow densities were 206 kg m$^{-3}$, 337 kg m$^{-3}$, 328 kg m$^{-3}$ in 2016, 2017 and 2018, respectively.



## 3       Data and Methodology

### 3.1       Terminology

Assessing and comparing simulations to observations of ice cover can be challenging due to the differing definitions used, this

paper will follow definitions determined by Brown and Duguay (2011a). The date when simulations form a permanent/complete ice cover for the season is defined as ice-on, with the first day of simulated open water defined as ice-off. The ice-on/off date is defined as the first day when ice/open water is detected above the ice thickness sensor. The camera imagery is subjective since visual assessment and interpretation of ice conditions are impacted by light availability and weather conditions, however, images can be used to identify the freeze-up period, which is defined as the time between when the ice

is initially visible in the camera view (freeze-onset) until the formation of solid ice cover (complete ice-on). Surface ice decay can also be identified with camera imagery and is defined by the time when any ice in the camera view is visibly beginning to melt (snow free, wet/slushy surface). The break-up period is defined as the date between the first appearance of open water and when water is completely free of ice (complete ice-off). Dates extracted from the CID represent complete freeze over (ice-on) and when the water body is clear of ice (ice-off) (Lenormand et al., 2002).

### 3.2       Snow and Ice Measurements

#### 3.2.1       High Arctic

Intense data collection is not logistically possible for the High Arctic lakes, however in situ measurements of snow depth and density were measured through end-of-season snow survey's (Woo, 1997) on Small Lake for May 20, 2016 and May 16, 2018. Sampling transects ranged from 200 to 700 m, snow depth was measured every 10 m along each transect, with density

measurements taken at the start, mid-point and end of each transect. To monitor ice conditions and snow redistribution -40°C-rated outdoor digital trail cameras (RECONYX PC800 HyperFire Covert) were installed at each lake. One camera was installed at Small Lake prior to ice-off in May 2016 and a second was installed prior to ice-on at Resolute Lake (Fig. 1) in August 2017. The cameras are in locations selected to maximize field of view of the lake and to allow for accessibility; they capture daily imagery at mid-day to maximize daylight conditions later in the season. No camera imagery was collected for ice-off at Small

Lake in 2017 due to a camera power issue. To extend the ice cover record for Resolute Lake, available ice-on and ice-off dates were obtained from the CID from 1960 to 1985. In addition, 1 km Moderate Resolution Imagining Spectroradiometer (MODIS) corrected reflectance (true colour) images from Worldview were used to visually estimate ice-on and ice-off dates between 2000 to 2017 when no observed ice dates are available.

#### 3.2.2       Temperate

Building from the Ariano and Brown (2019) study, in situ measurements of ice thickness, ice composition, snow depth, and snow density were recorded weekly when ice conditions permitted for the 2015–2016, 2016–2017 and 2017–2018 field seasons. Four sampling transects (Fig. 1) ranging from 50 – 400 m were established, with two on each study lake; snow depth





and snow density were measured following the same protocol as on Small Lake. In addition, manual ice thickness measurements were taken along each transect noting the total ice thickness, and the thickness of black and white ice layers. Ice thickness and layer composition were averaged weekly for both each study lake.

Six RECONYX cameras similar to the ones used in the High Artic were installed on trees around the study lakes. Placement of the cameras was selected based on road accessibility and for maximizing the field of view for capturing the snow and ice conditions of the centres and bays of the lakes on an hourly basis (Fig. 1). Continuous ice evolution was monitored using a Shallow Ice Water Profiler (SWIP, ASL Environmental) for the 2016–2017 and 2017–2018 study years, deployed at a depth of ~ 3m in MacDonald Lake, within the field of view of the on-shore AWS for data validation purposes. The SWIP was monitoring the ice thickness every 2 min and the data were extracted and processed following the similar protocol outlined previously by others (e.g. Melling et al., 1995, Marko et al., 2006, Brown and Duguay 2011a, Ariano and Brown, 2019).

**3.3 Albedo**

Detailed albedo was measured by Heron and Woo (1994) and these values form the basis of the melt parameterization currently used in CLIMo (Duguay et al., 2003). For the temperate lakes, surface albedo of the lake ice and on-ice snow were measured on each snow transect (start, middle and end) using a hand-held Solarmeter® Model 10.0 Global Power Meter and averaged for each week (end of the 2018 and throughout the 2019 field campaigns). To obtain the ice albedo, the snow was cleared away from the surface of the ice. During two separate site visits, the ice was snow-free which allowed us to obtain ice surface albedo only (Fig. 2; February 23 and March 2). In addition to the hand-held measurements, a Kipp and Zonen CNR4 net radiometer (measuring downward and upward facing solar radiation in the 0.3 to 2.8 μm range every 60 seconds), was mounted and levelled, extended from a dock 1.2 m above the snow/ice surface on MacDonald Lake (within the 1–2 m above-surface range indicated as ideal by the World Meteorological Organization (WMO, 2008)) for a full season of 8 dates in 2019. Albedo values were calculated by dividing the total reflected shortwave radiation by the total incoming shortwave radiation during daylight hours.

**3.4    Lake Ice Model**

A full description of CLIMo can be found in Duguay et al. (2003), while this abridged description follows our own papers; Brown and Duguay (2011a), Brown and Duguay (2011b), Svacina et al. (2014), and Gunn et al. (2015). CLIMo has been adapted from the one-dimensional thermodynamic sea-ice model of Flato and Brown (1996) which is based on the one-dimensional unsteady heat conduction equation, with penetrating solar radiation, of Maykut and Untersteiner (1971):

$$\rho C_p \frac{\partial T}{\partial t} = \frac{\partial}{\partial z} k \frac{\partial T}{\partial z} + F_{sw} I_0 (1 - \alpha) K e^{-K} \tag{1}$$





Where Duguay et al. (2003) define the following variables as: $\rho$ (kg m$^{-3}$) is the density, the specific heat capacity is $C_p$ (J kg$^{-1}$ K$^{-1}$), T (K) is the temperature (T(z, t) is the temperature within the ice or snow, t is time (s) and z is depth measured positive downward (m) from the upper surface), t (s) is the time, $k$ (Wm$^{-1}$ K$^{-1}$) is the thermal conductivity, $F_{sw}$ (Wm$^{-2}$) is the downwelling

shortwave radiative energy flux that penetrates the surface, $I_0$ is the fraction of shortwave flux that penetrates the surface (equal to 0.17 if snow depth is ≤ 0.01 m and equal to 0 if snow depth > 0.1 m), $\alpha$ is the surface albedo and K is the bulk extinction coefficient for penetrating shortwave radiation (m$^{-1}$).

To determine the net heat flux absorbed at the surface, the surface energy budget is calculated using:

$$F_0 = F_{lw} - \varepsilon \sigma T_s^4(0, t) + (1 - \alpha)(1 - I_0)F_{sw} + F_{lat} + F_{sens} \qquad (2)$$

where F$_0$ (Wm$^{-2}$) is the net downward heat flux absorbed at the surface, ε is the surface emissivity, σ is the Stefan-Boltzmann constant (5.67 x 10$^{-8}$ Wm$^{-2}$ K$^{-4}$), F$_{lw}$ (Wm$^{-2}$) is the downwelling longwave radiative flux, F$_{sw}$ (Wm$^{-2}$) is the downwelling shortwave radiative flux, F$_{lat}$ (Wm$^{-2}$) is the latent heat flux and F$_{sens}$ (Wm$^{-2}$) is the sensible heat flux (Duguay et al., 2003). The downward longwave energy flux is calculated using the formula of Maykut and Church (1973), F$_{lat}$ (Wm$^{-2}$) and F$_{sens}$ (Wm$^{-2}$) are the latent heat flux and sensible heat flux, respectively (both are positive downward) (Flato and Brown, 1996; Ménard et

al., 2002; Morris et al., 2005; Brown and Duguay, 2011a).

### 3.4.1 Albedo parameterization

An important component of the surface energy balance calculation used in CLIMo is albedo and it is used to determine ice melt and ice-off dates. As discussed earlier, CLIMo's surface albedo parameterization considers surface type, surface temperature and ice thickness. The surface albedo parameterization is summarized by Duguay et al. (2003) and Svacina et al.

(2014a) as:

$$\alpha = \begin{cases} \alpha_{ow} & h_i < h_{\min} \\ \min\left(\alpha_s, \alpha_i + \frac{h(\alpha_s - \alpha_i)}{0.1\,m}\right) & \begin{array}{l} h_i \geq h_{\min} \\ h_i \geq h_{\min} \end{array} & \begin{array}{l} h_i \leq 0.1\,m \\ h_s > 0.1\,m \end{array} \\ \alpha_s & h_i \geq h_{\min} \end{cases} , \qquad (3)$$

$$\alpha_i = \begin{cases} \max(\alpha_{ow}, 0.44\,m^{-0.28}\,h_i^{0.28} + 0.08) & T(0, t) < T_m \\ \min(\alpha_{mi}, 0.075\,m^{-2}\,h_i^2 + 0.15) & T(0, t) = T_m \end{cases} , \qquad (4)$$

$$\alpha_s = \begin{cases} 0.75 & T(0, t) < T_m \\ 0.65 & T(0, t) = T_m \end{cases} \qquad (5)$$

where α is surface albedo, α$_{ow}$ is albedo of open water (0.05), α$_s$ is the albedo of snow, α$_{mi}$ is the albedo of melting ice (0.55),

α$_i$ is the ice albedo, $h_i$ is ice thickness (m), $h_{\min}$ is minimum ice thickness below which open water is assumed (0.001 m), $h_s$ is





snow thickness (m), T(0,t) is the temperature within ice or snow at the vertical coordinate 0 at the time t (s), $T_m$ is melting temperature at the surface (273.15 K); 0.1 m, 0.44 m$^{-0.28}$ and 0.075m$^{-2}$ are derived from various field observations of ice thickness and radiative flux, which were documented by Maykut (1982) for cold ice. Equation (3) uses ice and snow thickness to determine the albedo value, whereas Eqs. (4) and (5) determine the value of both snow and ice albedo. The strong albedo

dependence of young ice on thickness is approximated in Eq. (3) and this has been derived from the observations of Weller (1972) on the radiation balance over sea ice. The melting ice parameterization is based on Arctic lake ice observations from Heron and Woo (1994) (at Small Lake), which has a larger proportion of black ice and therefore, a lower albedo than what would be found at the white ice dominated temperate lakes (Ariano and Brown, 2019).

### 3.5    Simulations

The simulations for Resolute Lake were run from 1958 to 2018 to match the ice record length from the CID (1958–1990) and the digital camera record (2017–2018), whereas, the simulations for Small Lake were run from 2016 to 2018 to match the record from the digital camera imagery. The model was driven by daily meteorological data obtained from Environment and Climate Change Canada (ECCC) (Table 3). Improved ice cover simulations are produced when snow density values monitored over the season are used over a fixed average value for the cold season and melt (Brown and Duguay, 2011a). This approach

was followed here, with bi-weekly density obtained from the Snow CD (MSC, 2000). The May end-of-season mean on-ice snow density measured May 2016 and 2018 at Small Lake (307 kg m$^{-3}$) is similar to the late May (May 15–31) value in the Snow CD (303 kgm$^{-3}$), which lends confidence to using the historical snow density values for the entire simulation period. Snow redistribution is prominent for Arctic lakes, with highly variable snow depths common across the lake surface, typically with less snow on-ice than is measured on land at local weather stations (e.g. Woo et al., 1983; Yang and Woo, 1999; Woo

and Young, 2004). Comparing the end-of-season snow depths measured at the Resolute weather station to Small Lake in 2016 and 2018 on the same days shows that only 65% and 40% of the station snow depth was measured on the ice. Both snow surveys had mean snow depths below the mean annual snow depth of 21 cm with the on-ice snow depths in May 2016 and May 2018 having standard deviations of 11.8 cm and 21.5 cm (respectively); suggesting large snow depth variations across the lake. To represent snow redistribution throughout the season two snow accumulation scenarios were used: 50%, to represent

the average amount of snow cover on the ice (and align with previous research for these lakes, Brown and Duguay, 2011b), and 0% to represent the maximum redistribution possible.

The model simulations for MacDonald Lake and Clear Lake were driven by daily onshore meteorological data (Table 3) from an AWS, located at the southwestern end of MacDonald Lake (Fig. 1d). Cloud cover was obtained for 2015 to 2018 from 1 km MODIS satellite imagery (MOD08_D3: daily mean cloud fraction) as no nearby ECCC stations collect cloud cover

information. Snow density was represented by the actual snow conditions on site. While on-lake snow density is typically denser than that measured on land (Sturm & Liston, 2003), the Snow CD (MSC, 2000) density values were not representative of current on-lake snow processes (Table 2), so field-measured densities were used in the simulations. Snow redistribution across the lake surface was accounted for by the snow cover scenarios for both MacDonald Lake and Clear Lake, which  were





simulated by determining the mean snow redistribution percentage (the difference between snow accumulation on shore and

the measured on-lake snow depth). The scenarios used for 2015–2016, 2016–2017 and 2017–2018 are 47%, 19%, and 24%, respectively. Additionally, to be able to validate CLIMo's simulated ice thickness with the SWIP, the mixing depth was set to 3 m for MacDonald Lake (and 6 m for Clear Lake, since it is a slightly larger and deeper lake).

Finally, to better represent the temperate ice cover, albedo values in Eqs. (4) and (5) were adjusted using measured on-lake snow and ice albedo values from the 2017–2018 and 2018–2019 field seasons. Pre-melt ice albedo was determined

January – March 2019 (excluding February 1 and February 8 due to rain events), where the albedo ranged from 0.71 to 0.84 (Fig. 4). Pre-melt ice albedo was set to a constant value of 0.75 (average value) in lieu of non-melt in Eq (4), as testing showed the measured albedo of the temperate region white ice was much higher than the existing parameterization accounted for. For example, using a range of 10 – 50 cm ice thickness returns albedos in the range of only ~0.3 – 0.44 with the existing parameterization. During the melt period identified by Henneman and Stefan (1999) for a temperate lake in Minnesota USA,

including the days with fresh snowfalls, the average melt albedo was ~0.5 to 0.6, which is a similar range to our record during the melt season, where our average albedo was 0.56 in 2019 under a mix of melting and snow days (March 8, 15, 22, 29 and April 5, after this point we could not sample on the ice). As the melt parameterization in CLIMo is based on black ice, simulations were run using a fixed albedo of 0.56 in lieu of the melting ice albedo in Eq. (4) to represent our ice cover. The existing maximum melt albedo in CLIMo is 0.55 and simulations using our melt albedo value of 0.56 yielded virtually the

same simulation results as using the existing maximum melt albedo of 0.55 (same thickness and ice-off timing). Average snow albedo was determined from the 2019 field season between January - March 8, 2019 (excluding mid and late March due to substantial slushing events on the lakes). Multiple snowfalls occurred through the season, with fresh snow (within 1-3 days) on the ice most weeks. The average snow albedo was 0.82 and the effects of using the field-based value on the ice simulations were investigated by altering the albedo by +/- 0.03, half of the standard deviation. Increasing the snow albedo yielded better

ice-off dates, however, for the study year 2015–2016 the snow albedo was increased to 0.88 to better predict ice-off dates as this season had more early-season snow on the ice. Since the results were similar using 0.55 vs. 0.56, this research kept the existing value of 0.55 as our field data consists of only five records; more detailed work should be conducted on melt season albedo in temperate latitudes to build a larger sample size. In addition, pre-melt snow albedo in Eq. (5) was also adjusted to better capture the influence of more frequent fresh snowfalls (fresh snow was present during most site visits each field season)

that occur in temperate than in the Arctic (increased from 0.75 to either 0.85 or 0.88 based on snow conditions). We did not investigate altering the melting snow albedo beyond the existing parametrization in CLIMo (set deduction of 0.1 from the non-melt albedo), as only cold snow or slush were present on the ice during the sampling days. Similar to the melting ice albedo, further investigation into the albedo of melting on-ice snow in temperate regions should also be explored as a future research direction.


### 3.6    Model Performance

For datasets with more than 20 records, model performance was measured using the Index of Agreement in the R package 'HydroGOF' (Ia; standardized measure of the degree of model prediction error which varies between 0 and 1, where 1 indicates perfect agreement; Willmott, 1981; Zambrano-Bigiarini, 2017) and statistical errors were measured using Mean Bias Error (MBE; determines the systematic errors that occur and identifies if the values are being over- or under-estimated) and Mean Absolute Error (MAE; an absolute measure of the average magnitude of errors, with 0 indicating no error) in the R package 'tdr'(Lamigueiro, 2018). For smaller datasets, only statistical errors could be assessed using MAE.

## 4    Results and Discussion

### 4.1    Model Simulations – High Arctic

Simulations for ice-on at Small Lake correctly captured the observed first presence of ice for 2016 (September 12) and was within 1 day of the observed ice for 2017 (September 8, 2017; Fig. 3, Table 4 & 5).  The initial ice cover that formed was subsequently broken-up by wave action (visible in the camera imagery), resulting in the final ice-on date occurring 14 days later in 2016 and 4 days later in 2017. High wind speeds are often recorded in Resolute. CLIMo simulations return thermodynamic ice formation dates, hence the wind driven break-up events in this region were not captured (wind speed is used for the bulk formulae in the surface energy budget determination).

Ice-off simulations are heavily dependent on snow cover and a range of dates are framed by the two simulations. Simulations for 2016 showed ice-off on July 19 and 27 for 50% snow and no snow respectively, with observations indicating ice free conditions by July 21. Ice-off was simulated slightly later than observations in 2017, August 8 and 15, with observed ice-free conditions by August 1. The 2018 ice-off simulations indicated ice-free on July 27 and August 10, with the observed ice-free conditions occurring August 6. Open water can be observed in the camera imagery near the lake edges in mid-June as snowmelt runoff pools on the ice at the shore, forming a moat, and initiating near-shore melt. However, large floating ice pans can persist until late July or early August; on Small Lake the floating ice pans melted, and open water conditions coincide well with simulated ice-off.  For the three ice-off seasons simulated, the average error was 6 days (50% snow cover) and 8 days (no snow cover).

Resolute Lake (Fig. 4) also shows good agreement between observed and simulated complete ice-on dates (Table 6), with an Ia of 0.65 for 50% snow cover and 0.79 for no snow, a MBE of -3 and -4 days, respectively, which indicates a slight underprediction (earlier complete ice-on), and a MAE of 6 days for both snow cover simulations. Overall, the observed complete ice-on dates from the CID are modelled within 0 to 17 days of the observed complete ice-on dates, 0 to 7 days of the estimated MODIS imagery, and the camera imagery from 2017–2018 depicts complete ice-on within in 2 to 3 days of the simulated complete freeze-over. The MODIS imagery dates tend to slightly overestimate ice-on due to extensive cloud cover which obscures the ice processes, but the annual variability is in line with the model simulations.





The simulated ice-off dates for Resolute Lake using the 50% snow cover scenario are within 1 – 15 days of observations and within 26 days for the no snow scenario. The digital camera imagery in 2017 shows matching ice-off dates with the 50% scenario, while the no snow scenario, which would have grown thicker ice with no insulating snow overtop, simulates ice-off 7 days later. The estimated dates from the MODIS imagery are within 0 to 25 days and predominantly later than the simulations,

which is not unexpected as extensive cloud cover in the region can persist for consecutive days, obscuring the actual ice off date. Despite some years with large differences between the simulations and the observations, the results show a good year-to-year fit with an Ia of 0.75 for 50% snow cover and 0.77 for no snow cover scenarios. The ice-off for the 50% snow cover and no snow cover scenarios indicate an MBE of -2 days and 7 days respectively, and a MAE of 8 days for both scenarios (Table 6). These results show that the simulations vary between underprediction and over prediction, which is likely linked to

annual snow cover variability, and the occasional presence of residual ice pans. Resolute Lake is larger and deeper than Small Lake and can experience floating ice pans that remain later into the summer than on Small Lake, or in some cases persist through the summer and freeze into the new ice that forms in the fall. While ice pans were observed in both 2016 and 2017 (e.g. Fig. 5), records from the CID do not indicate the presence of residual ice pans which could lead to some discrepancy between the recorded ice-off dates and the simulations.

Using an average value of 50% snow cover to represent the long-term snow redistribution aims to represent a suitable amount of snow redistribution over the last 60 years. Using the snow survey data from the two available seasons (Table 1) on Small Lake can highlight the redistribution as evidenced through the large standard deviation recorded– particularly 2018 (11 cm mean, 21.5 cm SD). The range of on-ice snow depths (0 – 154 cm) indicates snow free in some regions and in other regions depth exceeding 100% of the on-land values in other down-wind (2018 only) and near-shore regions. The uncertainties in the

snow conditions of a given year attributed to redistribution are difficult to capture using one snow cover scenario and can result in the range of ice-off dates between snow cover scenarios.

Overall, the results of the simulations of ice-on and ice-off for both lakes, show agreement with previous studies from high latitude lakes where these studies simulated ice-on from 0 to 15 days and 1 to 10 days for ice-off, with the range affected by the snow cover scenarios (Ménard et al., 2002; Duguay et al., 2003; Jeffries et al., 2005; Brown and Duguay, 2011b).

**4.2    Unadjusted Model Simulations –Temperate**

Ice thickness and phenology were initially simulated using the unadjusted CLIMo for MacDonald and Clear Lakes using the snow depth differences (percentage of on-shore compared to on-lake snow depth) determined for each study year (Table 1) and the measured snow density (field density).The simulations are compared to observations for each year at two transects on each lake and provided in Fig. 6 (red line). Ice-on for MacDonald Lake was simulated well for the three seasons with an overall

MAE of 1 day compared to the camera imagery and 2 days compared to the SWIP (Table 7).  Ice-on at Clear Lake was not simulated as well, with a MAE of 10 days for the 3 seasons (Table 8). This larger error is suspected to be a result of the selected mixing depth representing deeper regions of the lake than where the cameras are capturing imagery in the shallower bays (where ice-on would likely occur a few days sooner).



Ice thickness is underpredicted (Fig. 6, Table 9) for both MacDonald and Clear Lake during 2015–2016, 2016–2017

and 2017–2018. Ice thickness for MacDonald Lake in comparison to the SWIP (Table 10) is 11.2 cm and 18.1 cm for 2016–

2017 and 2017–2018 respectively. In comparison to the observed thickness, MacDonald Lake has a mean MAE of 4.7 cm and

Clear Lake has a mean MAE of 8.4 cm for the entire study period (2016–2018). Spatial variability is evident when comparing

the manual measurements to the simulations and the SWIP, highlighting the thickness variability across the lakes. The overall

error was within 7 cm to the manual measurements, however, the end of season thickness was not included in the manual

measurements (e.g. Fig. 6, manual measurements ceased before maximum thickness) and the large discrepancy in the thickness

values becomes evident when compared with the SWIP (Fig. 6, Table 10). Furthermore, ice-off simulations were very poor

(Table 7 & 8) with overall errors approaching 4 weeks (25 – 28 days), clearly showing both ice thickness and ice-off dates are

not representative of these temperate lake sites.

The white ice formed in the temperate region presents a challenge within CLIMo with regards to adequately

simulating thickness throughout the ice-covered season, since the model does not currently include the contributions of

midwinter rain or meltwater refreeze on the ice. The current black ice (Arctic-based) parameterization also contributed to

underpredicting ice-off dates because the expected black ice (versus the actual white ice) has a lower albedo, which results in

a more rapid melt once underway. Therefore, to adequately represent ice thickness and melt simulations in the temperate region

with CLIMo, the albedo needs to be parameterized using field data that is representative of temperate lakes.

**4.3    Adjusted Model Simulations – Temperate**

Initially, changes were only made to the albedo of the ice during melt to explore the relationship between the white ice and the

melt rate and ice-off timing. While the results improved, this adjustment did not fully address the underprediction of thickness

and ice-off timing. Therefore, further simulations were run using an adjusted albedo parameterization of both snow and ice

(Supplementary Fig. S1 & Fig. 6; dark blue line), with the ice albedo being the dominant driver of the improvements. These

results were compared to the original unadjusted simulations.

Tables 7 through 10 highlight the substantial improvement to the model fit for the two temperate lakes. For

MacDonald Lake ice-off improved greatly from a MAE of 25 (SWIP) and 28 (digital camera) days to 0.5 and 4 days. Clear

Lake still shows some variation in the ice-on results, ranging from 1 to 13 days (MAE = 7, Table 8), however ice-off improved

substantially from a MAE of 30 days to 2 days. Simulated thickness improved substantially, with MAE values ranging from

2.0 – 4.5 cm across both lakes.

Due to unusually high snow depths in early in the 2015–2016 season compared to the other seasons (Ariano and

Brown, 2019), the snow albedo was set to 0.88 for this season to better represent both deeper earlier season snow that

accumulated on the ice and the multiple fresh snowfall events that occurred in mid-February and early March. For MacDonald

Lake (Fig. 6), model improvements simulated ice-on January 4th, which is the same date as observed. Ice thickness throughout

these field season fell between observed ice thickness data points which indicates a good agreement of the model to the

observed data. The MAE for ice thickness improved from 6.7 cm to 3.0 cm to 3.5 cm and 1.4 cm respectively for each transect



on MacDonald Lake (Table 9). With regards to Clear Lake (2015-16), ice thickness simulations showed a reduce average error, from 8.6 cm to 4.8 cm and 7.2 cm to 3.1 cm for the two transects. Some variation in the MAE for ice thickness is to be expected, as it would be attributed to spatial variability across the lake (from on-ice snow or bathometry variations), which would not be

captured by the 1-D model. For comparison, using the snow albedo of 0.85 to be consistent with the other seasons yielded an increased MAE of 1 to 1.5 cm and an earlier ice-off date of 5 to 7 days.

Model adjustments for 2016–2017 and 2017–2018 (snow albedo = 0.85) show a marked improvement for ice thickness compared to the SWIP data, with the Ia increasing to 0.98 and 0.92 (Table 10) in 2016–2017 and 2017–2018 respectively, with the end of season thickness now well represented. The ice thickness when compared to the transects shows

a MAE of 3.7 cm and 4.9 cm in 2016–2017 and slightly thicker at 6.8 and 6.1 cm in 2017–2018. The increase in thickness could be attributed to increased ice growth in the simulations due to colder than normal winter temperatures and less snow cover in January and late February. With regards to Clear Lake, the MAE improved to 2 cm or less for the 2016–2017 and 2017–2018 seasons (Table 9). In 2017–2018 the winter was initially colder and there were several warming events (with periods of rain) through the season. The adjusted simulation for MacDonald Lake shows good agreement with ice-on and ice-

off dates but ice thickness is slightly over-predicted compared to the SWIP (MBE of 1.9) and the observed field measurements (MAE of 6.8 for M1 and 6.1 for M2) are attributed to the spatial variation of ice thickness, as well as the colder temperatures and reduced snow cover (<5 cm) early in the winter season. Additionally, the simulations did not capture the increased thickness at the end of season to > 65 cm. The end of season cumulative rainfall for the month of March (69 mm) and runoff from the 11.4 km2 catchment, may have contributed to an increase in observed ice growth monitored in 2018 by the SWIP

when the weight of the wet snow depressed the ice cover. However, after this brief thickening, the melt rate and timing are very similar between the simulations and the SWIP.

The unadjusted model results indicate earlier ice-off dates, which we attribute to the lower albedo of black ice parameterized in the model, and hence does not account for the delay in ice melt attributed to the predominantly white ice. This supports the work by Ashton (1986) which states that snow-ice (white ice) slows the ice thinning rate during the early

melt season. By utilizing field-based parameterization values of ice and snow albedos in the adjusted simulations, ice decay and ice-off timing are substantially delayed in the ice-cover season, resulting in much better representation of temperate region ice.

## 5 Summary and Conclusion

The results demonstrate the relationship between snow and ice composition on the simulation of lake ice formation, growth,
and decay of both High Arctic and temperate region lake ice using the Canadian Lake Ice Model. The High Arctic sites show good agreement to ice phenology dates, with a MAE of 6 days for ice-on and 8 days for ice-off for both snow cover scenarios (thickness could not be assessed since it was not recorded). Initial CLIMo simulations of two temperate lakes indicated ice-on dates within 1 to 10 days and ice-off dates of 25 to 30 days. Ice thickness was underpredicted by up to 13.7 cm on MacDonald



Lake. The initial results highlighted issues with the representation of the melt period within the model for temperate regions
where more white ice is present (Ariano and Brown, 2019). Adjustments to CLIMo used field measured albedo values for
snow and ice on temperate lakes and provided dramatically improved simulations of ice thickness and ice-off dates. The
adjusted simulations for ice-on had a MAE within 1 to 7 days and ice-off had a MAE of 1 to 4 days. Simulated ice thickness
over the 2016–2017 and 2017–2018 seasons improved from Ia of 0.62 to 0.98 and 0.50 to 0.92 respectively and was within
2.7 to 7.2 cm of the SWIP (MAE). Ice thickness compared to observations for all three seasons had a MAE within 2.0 to 4.5
cm. The albedo of temperate region snow and ice were all increased to better represent the frequent fresh snowfalls and large
amounts of white ice that are found in the temperate region. The higher albedo values reflect more incoming radiation, which
reduces the absorption of solar radiation into the ice cover and delays the simulated melt onset by approximately 1 month
which produces much more realistic results for this region. Overall, this research found that the surface albedo is critical to
represent correctly at temperate latitudes because of the impacts ice thickness, the timing of melt onset, and the final ice-off
dates. However, further investigation should also be completed regarding the effects of ice thickness on albedo, especially in
regions where white ice is dominant. It is important to understand how ice characteristics and cover are changing in temperate
latitudes, since freshwater lake abundance is highest within these latitudes (Verpoorter et al., 2014) and temperature projections
suggest the number of lakes transitioning from annual to intermittent winter ice cover will increase exponentially with climate
warming (Sharma et al., 2019). The ability to model lake ice cover with greater accuracy, including the correct representation
of the ice column, is a large stride towards a better understanding the feedbacks between lake ice to climate, in the years to
come.

**Data Availability.** The data that support the findings of this study are available from the corresponding author (ALR) upon
reasonable request.


**Funding information**. Canada Foundation for Innovation / Ontario Research Funds (Brown), Grant/ Award 34864; NSERC
Discovery Grant (Brown), Grant/Award Number: 5316; UTM Department of Geography Graduate Expansion Funds
(Robinson/Ariano); Northern Scientific Training Program (NSTP) (Robinson, 2017, 2018; Ariano, 2016); Haliburton Forest
and Wild Life Reserve Ltd.(in kind) and Polar Continental Shelf Program (PCSP) (logistical support, in kind).


**Author contributions.** AR and LB designed the study, SA and AR carried out the principle field component with guidance
and supervision from LB. AR and SA were involved in data curation and formal analysis including running preliminary model
simulations for the High Arctic and Temperate study site, respectively. AR modified the model code using in-situ field data
for the adjusted Temperate lake simulations. AR prepared the primary manuscript with contributions from all co-authors. LB
and AR were also responsible for visualization of data for publication.

**Competing interests.** The authors declare that they have no conflict of interest.



**Acknowledgements.** This project was funded by an NSERC Discovery Grant (Brown), CFI/ORF (Brown), UTM Department
of Geography Graduate Expansion Funds (Robinson/Ariano), Northern Scientific Training Program (Robinson/Ariano) and
the Polar Continental Shelf Program (Brown). We would like to thank Haliburton Forest and Wild Life Reserve Ltd. for their
in-kind support and overall assistance with this project. We would also like to thank Kathy Young, Scott Lamoureux, Laura
Thomson, Anna Pienkowski-Furze, Dana Stephenson, Debbie Iqaluk, Sean Yokoyama, Evan Thompson, Justin Murfitt, Alicia
Dauginis, William Sturch and the Qarmartalik School for field assistance. We also appreciate the assistance / advice from ASL
Environmental and Mike Brady (Environment and Climate Change Canada).

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

MacDonald and Clear Lakes weekly snow, ice and albedo sampling

**Figure 1. Location of field sites in Canada; High Arctic field sites (a) with zoom in (b) Small Lake and (c) Resolute Lake, Cornwallis Island, Nunavut; temperate field sites (d) with zoom in (e) MacDonald Lake and Clear Lake, Ontario. MacDonald Lake Automated Weather Station noted with Black circle at M2; Shallow Water Ice Profiler (SWIP) is also located at M2. Base map and lake boundaries © Statistics Canada (2016) and Canadian digital elevation model © Natural Resouces Canada (2015).**




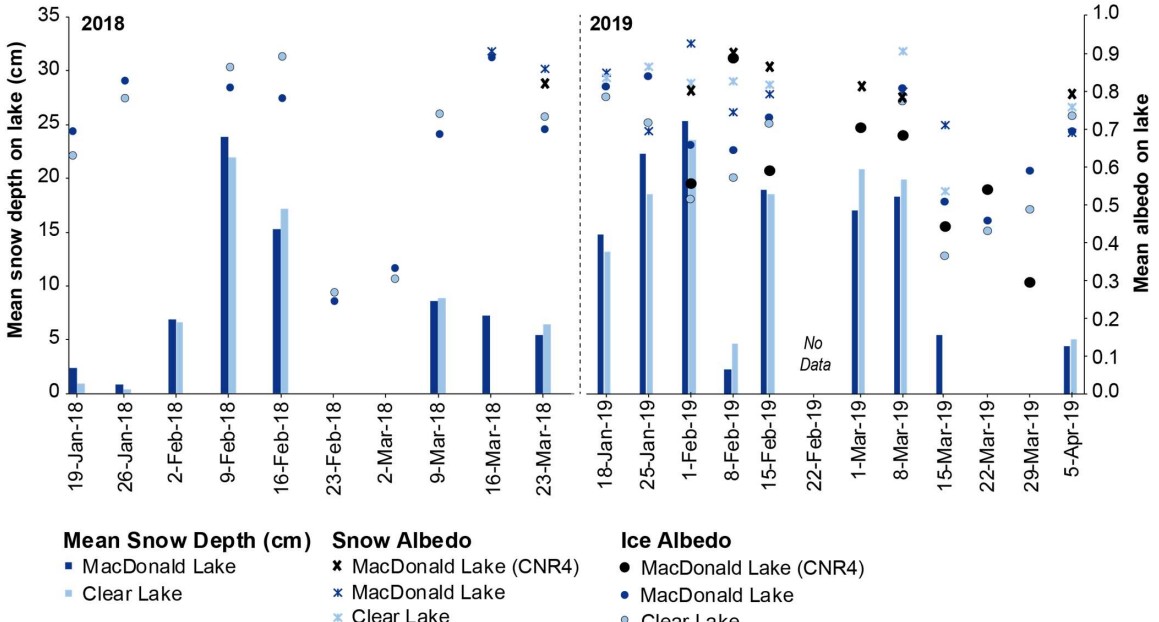

**Figure 2. Point measurements of lake ice albedo (2018 and 2019), on-ice snow albedo (2019), ice and snow albedo from the CNR4 (2019) and mean transect snow depths (2018 and 2019) from MacDonald Lake and Clear Lake.**


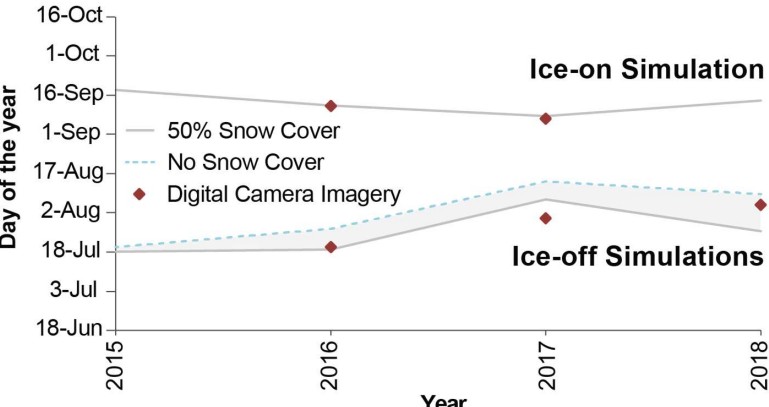

**Figure 3. Simulated ice-on and ice-off dates between 2015 and 2018 for Small Lake, NU compared with the observed ice-on and ice-off dates from the digital camera 2016-2018.**




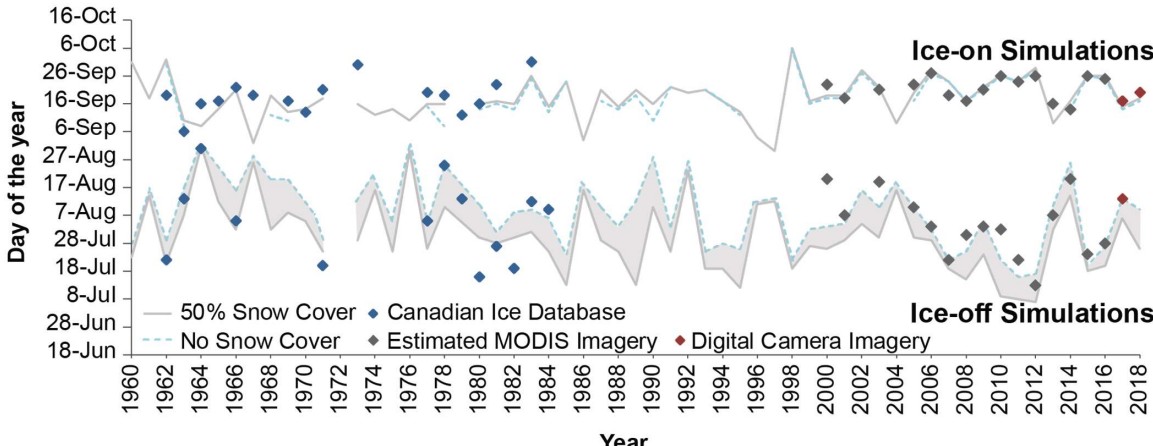

**Figure 4. Comparison between simulated ice-on and ice-off dates and those observed from the CID (1961–1986) and digital camera (2017–2018) for Resolute Lake, NU. Note, no ice-off (and therefore no ice-on) was simulated for 1972 with either scenario. Estimated ice-on and ice-off dates from MODIS imagery (2000–2018) are added for visual comparison but are not included in subsequent statistics as cloud cover results in substantial uncertainty.**

**Figure 5. Photograph of Resolute Lake taken August 8, 2018, labelled to show the difference between open water area and the ice pan at the north end of the lake.**





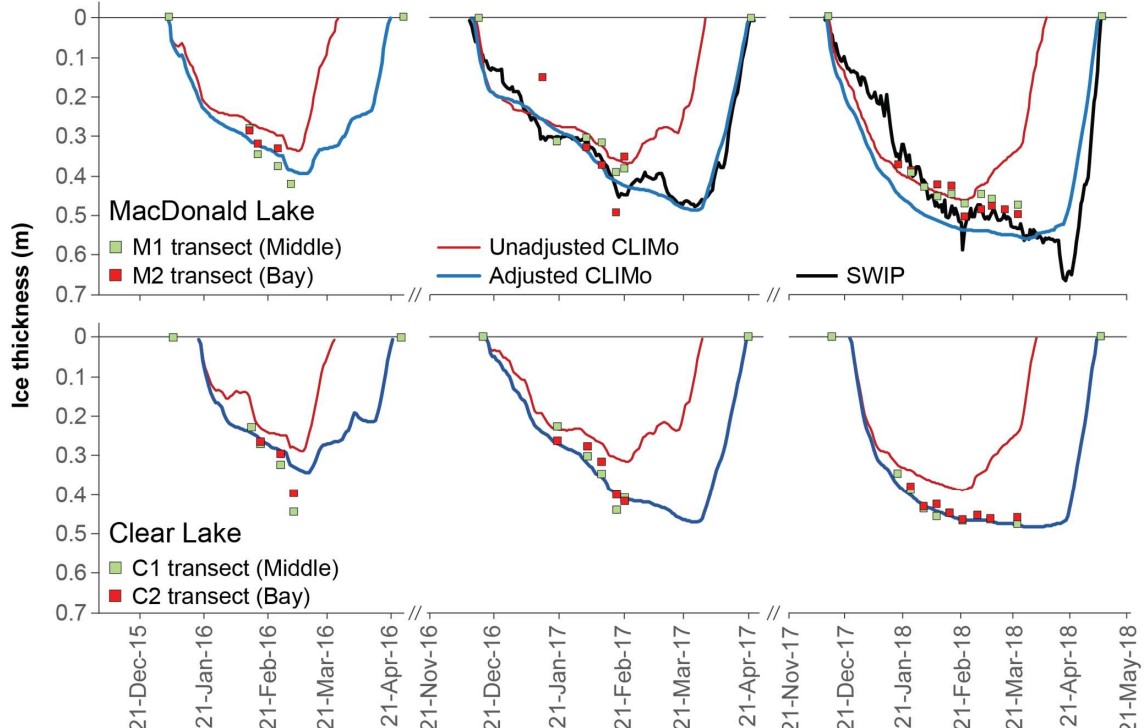

**Figure 6. MacDonald and Clear Lake model runs for three consecutive seasons showing the unadjusted model (red line) and adjusted model (dark blue line), compared to the Shallow Water Ice Profiler (SWIP, black line), and manual auger measurements (M-1: MacDonald Lake transect, middle of lake; M-2: MacDonald Lake, over the SWIP; C-1: Clear Lake, middle of Lake and C-2: Clear Lake, sheltered bay).**

**Tables**

**Table 1. End of season snow survey mean (standard deviation; SD) of snow depth (cm) and density (kg m³) for Small Lake, NU.**

| Year | | Snow Depth (cm) | Snow Density (kg m$^{-3}$) |
|------|------|------|------|
| **2016** | 22-May | 17.2 (11.8) | 356.5 (64.4) |
| **2018** | 16-May | 11.4 (21.5) | 307.7 (29.9) |





**Table 2. Average bi-weekly snow density (kg m³) from the Canadian Snow CD and weekly average of on-ice sampled snow depth (cm) and snow density (kg m⁻³) for 2015-2016, 2016-2017, and 2017-2018 for MacDonald Lake and Clear Lake.**

| Year | Day-Month | Snow Depth (cm) | Snow Density (kg m⁻³) | Corresponding Average Bi-weekly Canadian Snow CD density (kg m⁻³) |
|------|-----------|-----------------|------------------------|-------------------------------------------------------------------|
| **2016** | 22-Jan | 13.9 (1.3) | 196.2 (64.3) | 204 |
| | 12-Feb | 8.8 (2.4) | 183.5 (35.6) | 215 |
| | 16-Feb | 12.3 (0.8) | 165.3 (41.9) | 242 |
| | 26-Feb | 8.5 (2.1) | 305.2 (73.9) | 242 |
| | 04-Mar | 22.2 (2.0) | 193.5 (58.1) | 265 |
| **2017** | 13-Jan | 2.3 (1.2) | 376.6 (104.3) | 195 |
| | 20-Jan | 3.0 (0.9) | 424.7 (60.8) | 204 |
| | 03-Feb | 15.9 (1.2) | -- | 215 |
| | 10-Feb | 17.7 (1.7) | 371.2 (45.9) | 215 |
| | 17-Feb | 29.9 (4.2) | 206.5 (43.3) | 242 |
| | 21-Feb | 5.1 (4.1) | 307.4 (107.7) | 242 |
| **2018** | 19-Jan | -- | 297.7 (202.7) | 195 |
| | 26-Jan | 1.6 (1.0) | 102.0 (30.9) | 204 |
| | 02-Feb | 0.6 (0.4) | 87.6 (12.4) | 204 |
| | 09-Feb | 6.7 (0.2) | 163.3 (39.9) | 215 |
| | 16-Feb | 22.9 (1.6) | 109.8 (25.3) | 215 |
| | 23-Feb | 16.2 (1.6) | -- | 242 |
| | 09-Mar | -- | 120.8 (13.6) | 265 |
| | 16-Mar | 7.3 (1.7) | 450.3 (420.4) | 265 |
| | 23-Mar | 6.0 (0.8) | 190.7 (33.2) | 303 |






**Table 3. Data description for meteorological and snow data used for both study locations. Environment and Climate Change Canada (ECCC) data mainly used for the High Arctic sites and an on-shore Automatic Weather Station (AWS) primarily used for the temperate sites.**

| Variable | High Arctic | Temperate |
|---|---|---|
| **Air temperature** | ECCC Resolute CARS (1958-2014), Resolute Bay A (2014-2018) | On-shore AWS (2015 – 2018): HMP60 Temperature and Relative humidity probe |
| **Relative humidity** | ECCC Resolute CARS (1958-2014), Resolute Bay A (2014-2018) | On-shore AWS (2015 – 2018): HMP60 Temperature and Relative humidity probe |
| **Wind speed** | ECCC Resolute CARS (1958-2014), Resolute Bay A (2014-2018) | On-shore AWS (2015 – 2018): RM Young Wind Monitor |
| **Snow depth** | ECCC Resolute CARS (1958-2014), Resolute Bay A (2014-2018) | On-shore AWS (2015 – 2018): SR50A Sonic Ranging Sensor |
| **Cloud amount** | ECCC 1958-2018 | MODIS MOD08_D3: Cloud Fraction (Daily 1km product) |
| **Snow density** | ECCC: Snow CD  End-of-season snow survey's (Small Lake) May 22, 2016 & May 16, 2018 | ECCC: Snow CD  Field survey's weekly, 2016-2018 snow seasons |
| **OTHER** | | |
| **Solar radiation** | | CNR4 Net Radiometer (2018-2019)  Solarmeter® Model 10.0 Global Power Meter (weekly, 2018-2019) |
| **Barometric pressure** | | On-shore AWS (2015 – 2018): (CS106 Barometric Pressure Sensor) |






**Table 4. Comparison between simulated complete ice-on and ice-off dates and those observed from the digital camera at Small Lake, NU between 2016 and 2018.**

| Year | Ice-on (50% snow cover) | Ice-on (No Snow Cover) | Observed First Presence of Ice | Complete Ice-on | Ice-off (50% snow cover) | Ice-off (No Snow Cover) | Observed First Presence of Open Water (e.g. Ponding, moat formation) | Complete Ice-off |
|---|---|---|---|---|---|---|---|---|
| **2015-2016** | -- | -- | -- | -- | 19-Jul | 27-Jul | 17-Jun | 21-Jul |
| **2016-2017** | 12-Sep | 12-Sep | 12-Sep | 28-Sep | 8-Aug | 15-Aug | -- | 1-Aug |
| **2017-2018** | 9-Sep | 8-Sep | 8-Sep | 12-Sep | 27-Jul | 10-Aug | 16-Jun | 6-Aug |


**Table 5. Validation (Mean Absolute Error; MAE) results of simulated ice-on and ice-off dates compared to the first observed complete ice cover date and ice-off date from the digital camera at Small Lake, NU between 2016 and 2018.**

|  | 50 % Snow Cover – MAE | No Snow Cover - MAE |
|---|---|---|
| **Compete Ice-on** | 1 | 0 |
| **Ice-off** | 6 | 8 |

**Table 6. Validation results (Ia = Index of agreement, MAE= mean absolute error, MBE= mean bias error) of simulated complete ice-on and ice-off dates to the observed complete ice-on and ice-off dates from the CID from 1962 to 1994 and the observed date of first complete ice cover and ice-off from the digital camera at Resolute Lake, from 2017 to 2018.**

| Complete Ice-on | 50% Snow Cover | No Snow Cover |
|---|---|---|
| **$I_a$** | 0.65 | 0.79 |
| **MBE (days)** | -3 | -4 |
| **MAE (days)** | 6 | 6 |
| **Ice-off** | **50% Snow Cover** | **No Snow Cover** |
| **$I_a$** | 0.75 | 0.77 |
| **MBE (days)** | -2 | 7 |
| **MAE (days)** | 8 | 8 |




**Table 7. Observed and simulated ice-on/off dates for MacDonald Lake.**

| Study Year | Digital Camera | | SWIP | | Simulation: MacDonald Lake | | Adjusted Simulation: MacDonald Lake | |
|---|---|---|---|---|---|---|---|---|
| | Ice-On | Ice-off | Ice-On | Ice-off | Ice-On | Ice-off | Ice-On | Ice-off |
| **2015-2016** | 4-Jan | 27-Apr | | | 4-Jan | 26-Mar | 4-Jan | 20-Apr |
| **2016-2017** | 16-Dec | 22-Apr | 16-Dec | 21 Apr | 13-Dec | 31-Mar | 13-Dec | 20-Apr |
| **2017-2018** | 12-Dec | 8-May | 12-Dec | 7-May | 13-Dec | 8-Apr | 13-Dec | 6-May |
| | | | *MAE (days) digital camera* | | 1 | 28 | 1 | 4 |
| | | | *MAE (days) SWIP* | | 2 | 25 | 2 | 1 |



**Table 8. Observed, simulated and adjusted simulated ice-on and off dates for Clear Lake.**

| Study Year | Digital Camera | | Simulation: Clear Lake | | Adjusted Simulation: Clear Lake | |
|---|---|---|---|---|---|---|
| | Ice-On | Ice-off | Ice-On | Ice-off | Ice-On | Ice-off |
| **2015-2016** | 5-Jan | 24-Apr | 18-Jan | 21-Mar | 18-Jan | 19-Apr |
| **2016-2017** | 16-Dec | 20-Apr | 18-Dec | 29-Mar | 15-Dec | 19-Apr |
| **2017-2018** | 11-Dec | 8-May | 25-Dec | 11-Apr | 18-Dec | 6-May |
| *MAE (days) digital camera* | 10 | 30 | 7 | 2 |





**Table 9. Observed thickness versus unadjusted model thickness MAE in cm for study lakes M-1: MacDonald Lake transect, middle of lake; M-2: MacDonald Lake, over the SWIP; C-1: Clear Lake, middle of Lake and C-2: Clear Lake, sheltered bay.**

|  | *Unadjusted simulations* | | | | *Adjusted simulations* | | | |
|---|---|---|---|---|---|---|---|---|
| *Study Year* | **M-1** | **M-2** | **C-1** | **C-2** | **M-1** | **M-2** | **C-1** | **C-2** |
|  | **(cm)** | **(cm)** | **(cm)** | **(cm)** | **(cm)** | **(cm)** | **(cm)** | **(cm)** |
| **2015-2016** | 6.7 | 3.5 | 8.6 | 7.2 | 3.0 | 1.4 | 4.8 | 3.1 |
| **2016-2017** | 2.0 | 6.3 | 7.7 | 6.5 | 3.7 | 4.9 | 2.3 | 2.0 |
| **2017-2018** | 3.6 | 6.4 | 10.2 | 10.4 | 6.8 | 6.1 | 1.0 | 1.0 |
| *Overall* | 4.1 | 5.2 | 8.8 | 8.0 | 4.5 | 4.1 | 2.7 | 2.0 |

**Table 10. Model performance and error statistics for observed (SWIP) compared to simulated ice thickness for MacDonald Lake.**

|  | *CLIMo vs. SWIP* | | | *Adjusted CLIMo vs. SWIP* | | |
|---|---|---|---|---|---|---|
| *Study Year* | **Ia** | **MBE (cm)** | **MAE (cm)** | **Ia** | **MBE (cm)** | **MAE (cm)** |
| **2016-2017** | 0.62 | -9.9 | 11.2 | 0.98 | 2.0 | 2.7 |
| **2017-2018** | 0.50 | -13.7 | 18.1 | 0.92 | 1.9 | 7.2 |
