# Peer review of "The influence of albedo parameterization for improved lake ice simulation"

_Hydrology and Earth System Sciences, 2020_

## Referee Comment (RC1) · Anonymous Referee #1 · 29 May 2020

Review of the manuscript entitled " the influence of albedo parameterization for improved lake ice simulation " by Alexis L. Robinson, Sarah S. Ariano and Laura C. Brown.

Lake ice phenology is a key physical parameter both as a signature of climate changes and as a driver of lake ecosystem changes. Yet, lake ice remains complicated to model as it requires to couple ice/snow optical and mechanical properties to thermodynamic principles. The authors showed that the performance of a well-established thermodynamic model for lake ice simulations can be significantly improved by local information on the albedo. While the importance of the albedo has been long identified as key parameters regulating the heat flux (Leppäranta, 1993 for instance), deterministic models still struggle to correctly reproduce optical properties of the ice. In that sense, this

contribution is very relevant.

I have however several issues with the present manuscript. I found this manuscript very hard to follow and identify how the main goal is achieved.

1) Model description and discussion

If I understood correctly the main objective of this work is to show that the model performs better when the temporal evolution of the local albedo information is provided. While this is not a fundamental surprise that local forcing conditions are always better than global or random boundary parameters, I recognize that it is important to quantify the sensitivity of a model to such a parameter. Yet, the way the model is presented (Eqs. 1-5) is very confusing. For instance, the boundary conditions (especially the lower one at the ice-water interface) of the model are not presented. Off course, such information can be found in previous publications but the manuscript should be self-explicit. Furthermore, the authors presented 3 related equations to parameterize the albedo. They discuss optimization but it is very hard to understand what was really done. Specifically, there is no parameters in the three equations provided. It is thereby very hard to follow the changes described in the results. I think that the way the work is presented makes the study difficult for other scientists to follow and finally reproduce the findings. I thereby recommend to the authors to rewrite the results and discussion and reformulate the set of equations and finally discuss how the parameterizations have been modified to improve model's skill so that readers can understand what is modified.

2) Lower boundary condition

I have not understood how the flux at the lower boundary was calculated. This heat flux is proportional to the water temperature, which will evolve from let's say 0.1°C to 4°C, that is a 40 times increase in the heat flux. Said differently, the heat flux will range from < ∼1 W/m2 to > 10 W/m2 (without daily cycles). How is this increase over time of heat flux from the lower boundary taken into account? This boundary condition and

its variability is not discussed. Without a proper quantification or at least discussion regarding the lower boundary, one may think that changes in albedo parameterization are actually also taking into account changes in heat flux at the lower boundary.

3)Ice-on prediction

I have also not understood how this model can be used to predict ice-on as discussed in the results and discussion section. How much heat must be extracted from the lake before reaching the condition T=0°C? What are the initial conditions being used? The modeling approach is adapted to ice thickness and ice-off but not to ice-on.

4)Albedo

I am curious why the authors did not assimilate the observed albedo using Ensemble Kalman Filters or any other approach.

I also would need some clarifications regarding how albedo is actually measured. Some publications show huge daily variability (due to solar angle, melting and consequently daily changes in ice properties). It seems that you focus mostly on the seasonal variability that is the change from snow covered to snow free and further increase of the scattering with ice warming. I am curious if the daily variability is relevant or not.

Finally, it is mentioned many times that albedo drives the melting. This is potentially misleading. Solar radiation (and air temperature) drive(s) the melting. The albedo modulates the intensity of the forcing (see for instance a very cloudy day with -20°C but with very small albedo vs sunny day with +10°C with larger albedo).

5)Model skill metric

I suggest to develop the model's skill metric as a function of the calibration parameters to show the effective improvement of the new model's version. The authors actually discuss this in section 3.6 Model Performance, but I don't see the indexes in a figure, or how the calibrated parameters achieved an optimal value. The metrics are just shown in Tables 6 and 10, so we need to assume that these are the maximum values

achieved, but what are the optimal calibrated parameters?

6) Non exhaustive specific comments

*Abstract: first sentence not really attractive

*L10 "northern and temperate ice": odd formulation

*L28: why "exponentially"? More lakes up north?

*L43 "ice melt initiation is controlled by albedo". Please clearly stress that the drivers are solar radiation and air temperature.

*L45: Please distinguish main drivers to the secondary drivers for clarity reason

*Eq 3-5: Please add parameters instead of numeric values as you will be changing them. What is "m"?

*L246: "mixing depth" What is this? How many parameters do you have in the model? A table with the list of parameters with chosen values is needed. Very important for reproducibility reasons.

*L264 the standard deviation seems very small. Other studies report more that ~0.3 daily variability in albedo due to continuous changes of solar angle, cloudiness (scattering of downwelling radiation), and ice properties (melting, increase in the gas fractions) over the day.

*Figure 1. Latitude and longitude would help

*Figure 4. For model comparison purpose, the y axis is not ideal. I would split into 2 plots (and potentially remove ice on)

*Figure 6. Very nice figure. Maybe add the forcing (Tair, W) and the albedo time series used for "unadjusted" and "adjusted" cases. This is the central figure in my view. The performances of the "adjusted" model are impressive.

*Eq. (1). Is  a reference density or is it considered a dynamical variable?

---

## Referee Comment (RC2) · Anonymous Referee #2 · 20 Jun 2020

This paper describes the 1-D lake ice model simulations for the two northern lakes and temperate lakes in Canada with sensitivity studies by changing ice and snow albedo parameterizations. The paper also documents the recent observations of lake ice and snow properties in the two northern lakes and two mid-latitude lakes in Canada. This paper can potentially be a significant contribution to lake ice modeling in general. In particular, the observations on ice thickness, snow properties, and albedo are rare and valuable for model improvements. On the other hand, the paper appears to have issues with presenting their findings in a way that can contribute to the model's improvements, model description, and organization.

1) The manuscript does not feedback their findings well to the original albedo parameterization, which appears to fail to reproduce ice thickness and ice off dates in the

temperate lakes, but instead, it appears that the albedos were simply tuned to match the observed ice off dates and ice thickness in the temperate lakes. In addition, further tuning is done just for a specific season (2015-2016). The needs for these tunings indicate the opportunity of improvements in the original model but this has not been achieved in the way the work is presented.

Surprisingly, the fixed albedo values 0.75 (pre-melt) is equivalent for the albedo with 4.5m thick ice in eq. (3), and 0.56 (melting) is about the upper bound in eq. (4). The authors attribute the failure of the original parameterization to white ice in temperate lakes not taken into account by the model. This might be the case, but if so, this is a shortfall of the parameterization, which should actually be latitude-dependent, or include white ice. The model by Duguay (2003) appears to include snow ice (white ice) so including white ice in the albedo parameterization appears to be straight forward. The authors increased snow albedo just for the 2015-2016 season because of the massive snowfall earlier in the season, but doesn't this mean $\alpha_s$ in eq. (5) should be snow-depth dependent rather than altering $\alpha_s$ from year to year?

First, I suggest the authors provide a figure illustrating the relation among ice thickness, snow depth, surface temperature and bulk albedo for the original parameterization (yes it's multi-dimension but there can be a few ways for this, such as Figure 4 in Icepack Documentation, https://media.readthedocs.org/pdf/cice-consortium-icepack/icepack1.2.2/cice-consortium-icepack.pdf). Next, I suggest that the authors add data points of their observed albedos, as they have synchronized observations for snow depth, ice thickness, and albedos.

Finally, I suggest that the authors propose a new set of equations which includes white ice (ideally), or is latitude dependent (this could simply be another if branch). The improved parameterization would be a valuable contribution to lake ice modeling.

2) More details for the model description are needed. 'Mixing depth' appeared in section 3.5 but there is no description for what this does with the model. If the model

includes 'snow-ice' (or white ice) parameterization, this should be stated in section 3.4. How many layers are defined? Clarify that 'the vertical coordinate 0' means the interface between the air and the snow or ice? Is there any heat flux from water to the bottom of ice? I understand that many of them are described in Duguay et al (2003) but this paper should provide at least minimum of the key information.

3) The organization should be carefully reviewed. Multiple descriptions are mis-placed. For example, section 2 should be about geography and relevant background knowledge for the study areas but it extends to descriptions on CID and Snow CD (data used in this study), which should belong to section 3 "Data and Methodology". Adjustments to albedo in page 9 (section 3.5, simulation) should belong to section 3.4.1 (albedo parameterization).

Other points:

Page 1, L15: "the High Arctic ice cover" should be "ice cover in the High Arctic lakes"

Page 1, L17: The meaning of "underestimation" of "ice-off timing" is not clear.

Page 7, eq. (3): Please define 'h'.

Page 8, L211: Does 'the vertical coordinate 0' mean the air-ice/snow interface? How many layers were defined?

Page 9, L265: "However, for the study year 2015-2016 the snow albedo was increased to 0.88 to better predict ice-off dates as this season had more early-season snow on the ice". I don't understand this reasoning. Doesn't this simply mean the parameterization should include the increase of snow albedo with snow depth?

Page 11, L312: Please define 'Ia'.

Figure 1: It'd be helpful if bathymetry information for the lakes are added. No mean depth info for the High Arctic lakes? I see that they are provided for the temperate lakes in section 2.2.

Figure 6: I'd like to see thickness timeseries for the simulations in the High Arctic lakes as well. Do they capture the feature in the historical observations described in section 2.1?

More information on forcing (air temperature, wind, snowfall) would be appreciated. Maybe timeseries graphic or providing mean values for each season.

---

## Author Comment (AC1) · 28 Jul 2020

We thank the reviewer for their time and effort in providing a constructive evaluation of our study and for their comments regarding how to improve the readability/flow of the paper. Several of the main comments revolve around the model physics that we have not explained in depth in the manuscript. In the interest of space and relevance, we did not go into great depth in some areas as the model physics are well established and our aim was to show how field data can be used to improve the representation of the ice cover in different regions, not alter the established physics. The full model description is referenced (Duguay et al., 2003) and does address much of the information the Reviewer was raising. We have identified some areas of the manuscript, as highlighted by the Reviewer (and Reviewer #2), that we can add further clarification too and also

some areas that we can remove. Equations 1 and 2 are only included as general model description included for continuity with previously published CLIMo papers.

In general, CLIMo simulates a continuous time series, from ice formation, growth, decay, summer ice free season and into the subsequent years. Freeze up is simulated very accurately provided accurate climate and lake data is used to run the model. CLIMo does not require calibration or optimization. Our research presented here focusses on using field measurements to better describe the albedo parameterization for a specific location – temperate region where the ice is different than northern regions. We are not adjusting any of the model physics using the parameterization, only the fixed values used to represent the ice in the temperate region vs. northern regions.

Replies to each comment are provided below.

Reviewer: 1) Model description and discussion: If I understood correctly the main objective of this work is to show that the model performs better when the temporal evolution of the local albedo information is provided. While this is not a fundamental surprise that local forcing conditions are always better than global or random boundary parameters, I recognize that it is important to quantify the sensitivity of a model to such a parameter. Yet, the way the model is presented (Eqs. 1-5) is very confusing. For instance, the boundary conditions (especially the lower one at the ice-water interface) of the model are not presented. Off course, such information can be found in previous publications but the manuscript should be self-explicit. Furthermore, the authors presented 3 related equations to parameterize the albedo. They discuss optimization but it is very hard to understand what was really done. Specifically, there is no parameters in the three equations provided. It is thereby very hard to follow the changes described in the results. I think that the way the work is presented makes the study difficult for other scientists to follow and finally reproduce the findings. I thereby recommend to the authors to rewrite the results and discussion and reformulate the set of equations and finally discuss how the parameterizations have been modified to improve model's skill so that readers can understand what is modified.

Reply: To clarify – there is no temporal evolution of the albedo being used in this study. We are using an average value to better represent the whiter surface ice in the study region compared to typical ice found in northern latitudes. No equations have been modified; only the fixed values representing the snow and ice albedo were altered as an initial approach to better simulate the temperate ice cover.

The overall research compares ice cover simulations from High Arctic and temperate region lakes to illustrate the latitudinal differences in lake ice properties and presents refinements to CLIMo to better simulate ice thickness and ice-off timing in the temperate region. Where the first objective is to show the effectiveness of CLIMo for simulating ice cover regimes on High Arctic lakes, where no changes have been made to the model. The second object is to investigate and show that certain parameters within CLIMo, which were determined using High Arctic research, need to be adjusted to appropriately simulate temperate region ice cover. The results and discussion are presented in separate sections to clearly meet these objectives.

The methodology section however was written from the perspective of the unadjusted model (Section 3.4) with the modifications outlined in section 3.5. Equations 3 to 5 are important to explaining CLIMo for northern lakes (High latitude lakes), as our research supports previous studies (e.g. Ménard et al., 2002; Duguay et al., 2003; Morris et al., 2005; Brown and Duguay, 2011a, 2011b; Kheyrollah Pour et al., 2012; Surdu et al., 2014) which show that the current unadjusted CLIMo is able to model sub-Arctic and Arctic lakes (northern lakes). These equations define the current parameters and how they were derived, explaining why simulations of temperate region lakes show earlier ice-off and lower thickness estimates.

Rearranging the methodology section will greatly help clarify the modifications that were made. Currently, the changes are detailed through text in section (3.5 Simulations) but rearranging this section to relate the changes directly to the equations will aid greatly for readability and clarify the albedo differences. We see the reviewers point regarding fixed values vs. parameters, in equation 5 and thank them for bringing this

to our attention. In the current form the description can lead to confusion as the specific numbers that we alter are not indicated. We will modify this to use parameters for further clarity.

By combining and rearranging section 3.4 and 3.5 a much clearer description of the model and modifications can be presented. We will re-label section 3.4 to 'Unmodified Lake Ice Model – Northern Lakes' and state this section reviews the model as it was originally created for high latitude lakes; change 3.5 to 'Modified Lake Ice Model – Temperate Latitudes' and clearly outline here what and how the albedo values were obtained. Some relevant material from the discussion section can also be worked into this modified section to clearly show the reader the differences between the unadjusted and adjusted model. Some additional helpful advice from Reviewer #2 will also be factored into the rearranged sections here to better present the work.

Reviewer: 2) Lower boundary condition I have not understood how the flux at the lower boundary was calculated. This heat flux is proportional to the water temperature, which will evolve from let's say 0.1C to 4C, that is a 40 times increase in the heat flux. Said differently, the heat flux will range from < ~1 W/m2 to > 10 W/m2 (without daily cycles). How is this increase over time of heat flux from the lower boundary taken into account? This boundary condition and its variability is not discussed. Without a proper quantification or at least discussion regarding the lower boundary, one may think that changes in albedo parameterization are actually also taking into account changes in heat flux at the lower boundary.

Reply: The calculation of the lower boundary conditions is not discussed in this paper as they are not directly relevant to the focus of the paper, but they are referenced in other papers concerning CLIMo – primarily Duguay et al., 2003. The one-dimensional unsteady heat condition equation (Eq 1) is subject to lower boundary conditions at the ice/water interface (underside), this incorporates the total thickness of the ice and snow (h) and the freezing temperature of fresh water (however the ice underside is always at the freezing point; Duguay et al., 2003). Growth and melt at the ice/water interface (underside) are computed from the difference between the conductive heat flux into the ice and the heat flux out of the upper surface of the mixed layer. Therefore, ice thickness is incorporated into the lower boundary conditions, since it is used to determine the penetration of shortwave radiation through the bottom of the ice slab. When shortwave radiation penetrates through the bottom of the ice slab, it is assumed to be absorbed within the mixed layer and then returned to the ice underside in order to keep the temperature of the mixed layer at the freezing point (Duguay et al., 2003). We will revise the model description to provide more basics on the required information and direct the reader to the correct place for relevant but not critical information, incorporating advice from reviewer #2 as well on this.

Reviewer: 3)Ice-on prediction I have also not understood how this model can be used to predict ice-on as discussed in the results and discussion section. How much heat must be extracted from the lake before reaching the condition T=0C? What are the initial conditions being used? The modeling approach is adapted to ice thickness and ice-off but not to ice-on.

Reply: We have addressed this point in our opening section, however, briefly, CLIMo simulates the annual cycle of a lake. Ice-on does not require adaptation for temperate regions; it is not discussed in detail within the paper since simulations are within 0 to 2 days and the ice/snow albedo has no effect other than some carry-through effects on heat storage with respect to the previous season break-up.

Reviewer: 4)Albedo I am curious why the authors did not assimilate the observed albedo using Ensemble Kalman Filters or any other approach. I also would need some clarifications regarding how albedo is actually measured. Some publications show huge daily variability (due to solar angle, melting and consequently daily changes in ice properties). It seems that you focus mostly on the seasonal variability that is the change from snow covered to snow free and further increase of the scattering with ice warming. I am curious if the daily variability is relevant or not.

Reply: While ideally a temporal series of albedo would have been used to determine the average, this was not viable at our field site as the location is used heavily for recreation and the equipment cannot be left unattended. Albedo measurements as stated in Section 3.3 Albedo, were obtained manually when site visits were made once per week. Albedo measurements are outlined in Section 3.3 'Albedo'. In 2017-2018 and 2018 – 2019)handheld measurements were made between 10 am to 2 pm, 3 were taken on each lake transect which allowed for a total of 12 measurements per site visit. The last two site visits in 2017-18 continuous shortwave radiation readings were made at MacDonald Lake between 10 am to 2 pm using a Kipp and Zonen CNR4 net radiometer (set-up detail outlined in the manuscript) and this was continued for a full season of 8 dates in 2019. We are seeking single values to best represent the snow and ice. Daily variability was not high as readings were all collected during the 10 am – 2 pm window. Examining the limited continuous data in comparison to the point data shows that the albedo averages to similar values during the overlapping collection times. Our variability is low likely as a result of the small range in hours that we are at the field site. For the purpose of this study, it was not found that variability was relevant since we were seeking an average value of albedo for temperate snow and ice conditions.

Assimilation of the observed albedo using Ensemble Kalman Filter or another approach is not required within the model, and not viable or appropriate using the limited field data collected. The model iterates daily through the entire study period using daily averages of air temperature, windspeed, relative humidity, cloud fraction, snow density and snow accumulation. CLIMo requires set values for albedo of ice, snow, melting ice, and open water. The parameterization of albedo is not an input value that would require assimilation and iteration of changing albedo values, such as what is determined using Ensemble Kalman Filters. In addition, this filter is used for large-scale datasets, and although it has been used extensively in hydrology and atmospheric sciences (Andreadis and Lettenmaier, 2006; Houtekamer and Mitchell, 2005; Roth et al., 2017; Zhang et al., 2019), the data being used in this study is limited point data and not large-scale spa-
tial data. This study was not about using remote sensing or big data to parameterize the model, however, if you would like more detail regarding the use of big data, such as MODIS daily albedo a study completed by Svacina et al. (2014a, 2014b) provides comprehensive detail on simulated and satellite derived surface albedo of lake ice and use in CLIMo.

Reviewer: Finally, it is mentioned many times that albedo drives the melting. This is potentially misleading. Solar radiation (and air temperature) drive(s) the melting. The albedo modulates the intensity of the forcing (see for instance a very cloudy day with -20C but with very small albedo vs sunny day with +10C with larger albedo).

Reply: A valid point to raise with our terminology. In our case, we were approaching the phrase with the model in mind in such that when the albedo changes melt initiates. We will find the instances of this phrase and reword accordingly.

Reviewer: 5)Model skill metric I suggest to develop the model's skill metric as a function of the calibration parameters to show the effective improvement of the new model's version. The authors actually discuss this in section 3.6 Model Performance, but I don't see the indexes in a figure, or how the calibrated parameters achieved an optimal value. The metrics are just shown in Tables 6 and 10, so we need to assume that these are the maximum values achieved, but what are the optimal calibrated parameters?

Reply: No calibration is required for CLIMo, the model is forced with daily mean meteorological values and a 2 year spin-up period is used. The supplemental figure shows the effects of each individual modification made to the albedo values and the resulting ice simulations, however, as noted we did not provide the metrics for the model fit of each iteration. We will add the index of agreement between each modification to the in situ thickness, as well as move the supplemental figure into the main manuscript to clearly show the improvements.

R1C6 Non exhaustive specific comments *Abstract: first sentence not really attractive

Reply: Good point. We will concoct a catchier lead-in for the abstract!

*L10 "northern and temperate ice": odd formulation

Reply: This will be rephrased for clarity. We cannot define set latitudinal ranges how-ever as the 'northern' limit dips quite far south around the bottom of Hudson Bay in Canada, and the temperate region is quite different latitudinally in Europe than in North America.

*L28: why "exponentially"? More lakes up north?

Reply: Yes, this is related to distribution of lakes. Research from Verpoorter et al. (2014) show that the greatest lake abundance between 45-75° N and this is supported by the research completed by Prowse et al. (2015) where high resolution satellite im-agery was used to illustrate that the highest concentration of lakes by area and perime-ter are between 45-75° N. In recent work by Sharma et al. (2019) on lake ice loss in the Northern Hemisphere, found that the number of lakes experiencing intermittent winter ice cover is "projected to increase exponentially with climate warming (p. 3), and highlighted in figures within that study.

*L43 "ice melt initiation is controlled by albedo". Please clearly stress that the drivers are solar radiation and air temperature.

Reply: This will be rephrased to include information that the drivers of albedo are solar radiation and air temperature. "ice melt initiation is controlled by albedo, where the main drivers are solar radiation and air temperature. Albedo is a surface property..."

*L45: Please distinguish main drivers to the secondary drivers for clarity reason

Reply: This will be rephrased to distinguish the main and secondary drivers for reader clarity. "Lake ice albedo is primarily affected by snow cover, ice type (e.g. black ice and white ice) and ice thickness, but can also be affected by the presence of impurities, cloud cover, air temperature and solar zenith angle (Leppäranta, 2015)."

\*Eq 3-5: Please add parameters instead of numeric values as you will be changing them. What is "m"?

Reply: Tm is melting ice temperature; however, m (italics) alone is a typo and it should be m which is a reference to the unit m for meters. This m (italics) will be changed to m to rectify the error in the equation. Parameters will be used in

---

## Author Comment (AC2) · 28 Jul 2020

We thank the reviewer for their time and effort in providing a constructive evaluation of our study and particularly for their comments regarding a better organization and relevant model details. The manuscript will be strengthened and improved by rearranging the methods section in particular to clearly outline the improvements made. We will aim to modify the manuscript as suggested throughout. Responses are listed below each comment.

R2C1: The manuscript does not feedback their findings well to the original albedo parameterization, which appears to fail to reproduce ice thickness and ice off dates in the temperate lakes, but instead, it appears that the albedos were simply tuned to

[Figure]

match the observed ice off dates and ice thickness in the temperate lakes. In addition, further tuning is done just for a specific season (2015-2016). The needs for these tunings indicate the opportunity of improvements in the original model but this has not been achieved in the way the work is presented.

Reply: To clarify, the albedo was only tuned for 2015-2016 to better fit the ice on /off dates – the main value used for the other years was collected from field data and the average value was used. The standard deviation was used to further explore the effects of the unusual snow year in 2015-2016 (albedo of 0.88 vs. 0.85) to account for the early/frequent snowfalls in that unusual year. In 2015-16 the climate data shows that this winter was warmer and had earlier season snowfalls that exceed those in 2016-17 and 2017-18. However, no albedo was recorded in the 2015-16 season. This season had deeper earlier season snow (Table 2) and a lower snow density (Table 2) more indicative of fresh snow, which would have a higher albedo compared to the snow density values measured in 2016-17 and 2017-18 during the same January period. Therefore, our reasoning was to increase the snow albedo value to account for the fresher snow that was measured in 2015-16.

This was not explained well on our part. We will revise to clarify, in particular by moving the supplemental figure to the main manuscript to show clearly the minor difference between using the same albedo for all years, rather than fine tuning the 1 year with no albedo data collected. As suggested by Reviewer #1 we will also add the model metrics to the interim simulations leading up to the final adjustments to clearly show the effect of each individual albedo adjustment.

Reviewer: Surprisingly, the fixed albedo values 0.75 (pre-melt) is equivalent for the albedo with 4.5m thick ice in eq. (3), and 0.56 (melting) is about the upper bound in eq. (4). The authors attribute the failure of the original parameterization to white ice in temperate lakes not taken into account by the model. This might be the case, but if so, this is a shortfall of the parameterization, which should actually be latitude-dependent, or include white ice. The model by Duguay (2003) appears to include

snow ice (white ice) so including white ice in the albedo parameterization appears to be straight forward. The authors increased snow albedo just for the 2015-2016 season because of the massive snowfall earlier in the season, but doesn't this mean $\alpha$s in eq. (5) should be snow-depth dependent rather than altering $\alpha$s from year to year?

Reply: Potentially, yes, but there is not enough field data to conclusively quantify the snow albedo – snow depth relationship for the study area. There are no albedo data for the 2015-16 season, and the snow melts frequently in this region so continuous on-ice data would be needed. Snow depth on lake ice are very rare measurements to have in a dataset, so including this as a required value to run the model would limit the applications – the overall goal is an acceptable modification to be widely applicable to temperate regions. The 0.88 value was more meant to represent the large fresh snow than the depth per se. We will reword the relevant text to clarify that 0.88 was an exploration to see if the different snow conditions could be represented.

Reviewer: First, I suggest the authors provide a figure illustrating the relation among ice thickness, snow depth, surface temperature and bulk albedo for the original parameterization (yes it's multi-dimension but there can be a few ways for this, such as Figure 4 in Icepack Documentation, https://media.readthedocs.org/pdf/cice-consortiumicepack/icepack1.2.2/cice-consortium-icepack.pdf). Next, I suggest that the authors add data points of their observed albedos, as they have synchronized observations for snow depth, ice thickness, and albedos.

Reply: This is a great idea, thank you, we will explore this and work in a new figure to better explain the current parameterisation in conjunction with our field data. A new project is starting in our group this fall that focusses more heavily on the snow depth/ice thickness/albedo relationship. We have only limited data at the moment, but will ideally be able to present a more conclusive story of the relationships in the temperature regions after a few more years of data are added (currently 3 years, aiming for 2 more to capture more climate variability).
Reviewer: Finally, I suggest that the authors propose a new set of equations which includes white ice (ideally), or is latitude dependent (this could simply be another if branch). The improved parameterization would be a valuable contribution to lake ice modeling.

Reply: With respect to the snow ice currently parametrized in the model – this is based on the typical mass related slushing that would occur on northern lakes and does not capture melt/refreeze that occurs in the temperature regions (Ariano and Brown, 2019), hence we are not focussing on quantifying/validating the current snow ice parameterization. This is beyond the scope of this paper; however, it is the focus of an upcoming research project, aiming to quantify the white ice formation from the multiple mechanisms possible in the temperate regions. We can quantify how much white ice is present, but we cannot currently separate the formation mechanism – typical slushing or melt water refreeze and hence cannot parameterize it correctly yet. The end goal is to account for geographic location (temperate vs. northern) in the selection of which albedo values to use in the simulations. The current paper is the first exploration of adjusting the model for temperature regions to represent the overall ice thickness and timing. Future work will delve into the composition complexities.

R2C2: More details for the model description are needed. 'Mixing depth' appeared in section 3.5 but there is no description for what this does with the model. If the model includes 'snow-ice' (or white ice) parameterization, this should be stated in section 3.4. How many layers are defined? Clarify that 'the vertical coordinate 0' means the interface between the air and the snow or ice? Is there any heat flux from water to the bottom of ice? I understand that many of them are described in Duguay et al (2003) but this paper should provide at least minimum of the key information.

Reply: Thank you for highlighting the missing information. A fixed mixing depth is used to in CLIMo to represent the mixed layer depth. In CLIMO, when ice is present, the mixing depth layer is fixed at the freezing point, otherwise when ice is absent, the mixing layer temperature is computed from the surface energy budget (Duguay et al.,

2003). We will add a brief description of the mixing depth effects, layers and the heat flux. (Snow ice comment addressed above).

R2C3: The organization should be carefully reviewed. Multiple descriptions are misplaced. For example, section 2 should be about geography and relevant background knowledge for the study areas but it extends to descriptions on CID and Snow CD (data used in this study), which should belong to section 3 "Data and Methodology". Adjustments to albedo in page 9 (section 3.5, simulation) should belong to section 3.4.1 (albedo parameterization).

Reply: Thank you for your suggestions, we will revise as suggested while factoring in some reorganization suggested by Reviewer #1 as well. We will be dividing the methods section describing the albedo in CLIMo into 'unadjusted' and 'adjusted' to clearly outline the changes and field data collection.

Other points: Reviewer: Page 1, L15: "the High Arctic ice cover" should be "ice cover in the High Arctic lakes"

Reply: This will be revised to "Simulations of High Arctic lake ice cover..."

Reviewer: Page 1, L17: The meaning of "underestimation" of "ice-off timing" is not clear.

Reply: Underestimation of ice-off timing refers to simulated complete ice-off (break-up timing) occurring earlier than actual (from camera imagery and the SWIP) complete ice-off (water-clear of ice). This section will be reworded to match Section 3.1 terminology where break-up and ice-off are defined as complete ice-off.

Reviewer: Page 7, eq. (3): Please define 'h'.

Reply: This will be revised to "h is the total thickness of the snow and ice layers"

Reviewer: Page 8, L211: Does 'the vertical coordinate 0' mean the air-ice/snow interface? How many layers were defined?

Reply: the standard 5 layers used in CLIMo for previous research were held for these simulations as well. Where one layer represents the snow and four represent the ice. We will clearly outline this in the text where indicated.

Reviewer: Page 9, L265: "However, for the study year 2015-2016 the snow albedo was increased to 0.88 to better predict ice-off dates as this season had more early-season snow on the ice". I don't understand this reasoning. Doesn't this simply mean the parameterization should include the increase of snow albedo with snow depth?

Reply: (See above)

Reviewer: Page 11, L312: Please define 'Ia'.

Reply: This will be revised and added to Section 3.6 Model Performance Line 276: "...model performance was measured using the Index of Agreement (Ia) in the R package 'HydroGOF' (Ia; standardized measure of the degree of model prediction error which varies between 0 and 1, where 1 indicates perfect agreement; Willmott, 1981; Zambrano-Bigiarini, 2017)."

Reviewer: Figure 1: It'd be helpful if bathymetry information for the lakes are added. No mean depth info for the High Arctic lakes? I see that they are provided for the temperate lakes in section 2.2.

Reply: Since the model is representing the lake as a whole the bathymetry does not contribute greatly and adding the bathymetry would be a substantial undertaking as the maps are not available digitally. We will endeavour to include a reasonable estimate for the mean depth of the Arctic lakes based on the existing bathymetry maps from research work in the area, so that all four research lakes have mean and max depth provided.

Reviewer: Figure 6: I'd like to see thickness timeseries for the simulations in the High Arctic lakes as well. Do they capture the feature in the historical observations described in section 2.1?

[Figure]

Reply: Maximum ice thickness is provided for Resolute Lake between 1960 to 1984 in the Canadian Ice Database (CID; Lenormand et al., 2002), however no date is recorded for when the measurement was taken. For this reason, no ice thickness was included for Resolute Lake since we could not determine the accuracy of the daily thickness measurements for the model. With regards to Small Lake, no thickness measurements were recorded in the CID, however, we currently have a shallow water ice profiler (SWIP) deployed which is recording the full evolution of ice cover in this lake and plan to use this data for future comparison with the model to determine the accuracy of the ice thickness output. We will experiment with presentation ideas to include a dateless maximum thickness value to our thickness output, however, as mentioned there is no way to validate these simulations at this time.

Reviewer: More information on forcing (air temperature, wind, snowfall) would be appreciated. Maybe timeseries graphic or providing mean values for each season.

Reply: We will add the air temperature and snow data to the figure (similar to how we have done in Ariano and Brown, 2019) and explore the viability of adding albedo data as well. This should further highlight the benefits of the adjusted model as the climate link will be visually evident.

References:

Ariano, S. S., and Brown, L. C.: Ice processes on medium-sized north-temperate lakes, Hydrol. Process, 33, 2434– 2448, https://doi.org/10.1002/hyp.13481, 2019.

Duguay, C. R., Flato, G. M., Jeffries, M. O., Ménard, P., Morris, K., and Rouse, W. R.: Ice-cover variability on shallow lakes at high latitudes: model simulations and observations, Hydrol. Process, 17(17), 3465–3483, https://doi.org/10.1002/hyp.1394, 2003.

Lenormand, F., Duguay, C. R., and Gauthier, R.: Development of a historical ice database for the study of climate change in Canada, Hydrol. Process, 16, 3707–3722, https://doi.org/10.1002/hyp.1235, 2002.

Willmott, C. J.: On the validation of models, Phys. Geogr., 2, 184-194, https://doi.org/10.1080/02723646.1981.10642213, 1981.

Zambrano-Bigirarini, M.: hydroGOF: Goodness-of-fit functions for comparison of simulated and observed hydrological time series, https://cran.r-project.org/web/packages/hydroGOF/, 2017.
* * *

---

## Editor Comment (EC1) · Bettina Schaefli (Editor) · 6 Aug 2020

This manuscript presents an analysis of how a lake-ice model (CLIMo) simulates ice thickness and ice-off dates for two High arctic and two temperate lakes. Two reviewers asked for clarifications about the method and gave constructive feedback on the content of the paper, one recommended major revisions, one rejection.

The objectives of the study are summarized as (introduction): "This research compares ice cover simulations from High Arctic and temperate region lakes to illustrate the latitudinal differences in lake ice properties and presents refinements to CLIMo to better simulate ice thickness and ice-off timing in the temperate region."

As evident from the manuscript and the public discussion, the first part of the above

objectives is met in the sense that the paper discusses two different sets of lake simulations. The differences are summarized at the end of Section 4.2: "The white ice formed in the temperate region presents a challenge within CLIMo with regards to adequately simulating thickness throughout the ice-covered season, since the model does not currently include the contributions of midwinter rain or meltwater refreeze on the ice. The current black ice (Arctic-based) parameterization also contributed to underpredicting ice-off dates because the expected black ice (versus the actual white ice) has a lower albedo, which results in a more rapid melt once underway. Therefore, to adequately represent ice thickness and melt simulations in the temperate region with CLIMo, the albedo needs to be parameterized using field data that is representative of temperate lakes."

The second objective is in my view only partly reached: in fact, rather than refining the model (i.e. the model physics via its parametrization), the paper proposes to replace the default albedo (parameter) values by the average observed values.

The finding that the use of average observed albedo values leads to good model performances for two case studies is certainly valuable; or as stated in the public discussion, "The current paper is the first exploration of adjusting the model for temperature regions." In other words, this paper represents the application of an existing lake ice model to four case studies, with modified parameter values for the two of them. The model application and performance assessment required an important amount of observational data and data processing.

However, the overall result is in my view, not sufficient to justify a publication in HESS, whose scope is to publish "Research articles [that] report on original research which clearly advances our understanding of hydrological processes and systems, and/or their role in water resources management and Earth system functioning as detailed in the journal's aims and scope".
* * *
[Figure]

156, 2020.
Interactive
comment